# Inactivation of Material from SARS-CoV-2-Infected Primary Airway Epithelial Cell Cultures

**DOI:** 10.3390/mps4010007

**Published:** 2021-01-07

**Authors:** Kaitlyn A. Barrow, Lucille M. Rich, Elizabeth R. Vanderwall, Stephen R. Reeves, Jennifer A. Rathe, Maria P. White, Jason S. Debley

**Affiliations:** 1Center for Immunity and Immunotherapies, Seattle Children’s Research Institute, Seattle, WA 98101, USA; barrowk@uw.edu (K.A.B.); lucille.rich@seattlechildrens.org (L.M.R.); elizabeth.vanderwall@seattlechildrens.org (E.R.V.); stephen.reeves@seattlechildrens.org (S.R.R.); maria.white@seattlechildrens.org (M.P.W.); 2Department of Pediatrics, Division of Pulmonary and Sleep Medicine, Seattle Children’s Hospital, University of Washington, Seattle, WA 98101, USA; 3Department of Pediatrics, Division of Infectious Disease, Seattle Children’s Hospital, University of Washington, Seattle, WA 98101, USA; jennifer.rathe@seattlechildrens.org

**Keywords:** SARS-CoV-2, COVID-19, airway, epithelial, heat, protein, RNA, inactivation, Vero E6

## Abstract

Given that the airway epithelium is the initial site of infection, study of primary human airway epithelial cells (AEC) infected with severe acute respiratory syndrome coronavirus 2 (SARS-CoV-2) will be crucial to improved understanding of viral entry factors and innate immune responses to the virus. Centers for Disease Control and Prevention (CDC) guidance recommends work with live SARS-CoV-2 in cell culture be conducted in a Biosafety Level 3 (BSL-3) laboratory. To facilitate downstream assays of materials from experiments there is a need for validated protocols for SARS-CoV-2 inactivation to facilitate safe transfer of material out of a BSL-3 laboratory. We propagated stocks of SARS-CoV-2, then evaluated the effectiveness of heat (65 °C) or ultraviolet (UV) light inactivation. We infected differentiated human primary AECs with SARS-CoV-2, then tested protocols designed to inactivate SARS-CoV-2 in supernatant, protein isolate, RNA, and cells fixed for immunohistochemistry by exposing Vero E6 cells to materials isolated/treated using these protocols. Heating to 65 °C for 10 min or exposing to UV light fully inactivated SARS-CoV-2. Furthermore, we found in SARS-CoV-2-infected primary AEC cultures that treatment of supernatant with UV light, isolation of RNA with Trizol^®^, isolation of protein using a protocol including sodium dodecyl sulfate (SDS) 0.1% and Triton X100 1%, and fixation of AECs using 10% formalin and Triton X100 1%, each fully inactivated SARS-CoV-2.

## 1. Introduction

Severe acute respiratory syndrome coronavirus 2 (SARS-CoV-2) is rapidly infecting the human population, with approximately 80 million confirmed cases and over 1,740,000 deaths worldwide by late December 2020 [1]. While most cases of Coronavirus Disease 2019 (COVID-19) result in mild symptoms, some progress to respiratory and multi-organ failure [2,3,4], with a case fatality rate ranging from 0.2 to 27%, according to age group and medical comorbidities [5]. There is thus an urgent need to study host factors that impact the infectivity of SARS-CoV-2 in primary cell tissue culture, including at the level of the airway epithelium using live SARS-CoV-2 in vitro and ex vivo models. The U.S Centers for Disease Control and Prevention (CDC) issued guidance stating that work with live SARS-CoV-2 in cell culture, and initial characterization of material recovered from cultures infected with live SARS-CoV-2, should only be conducted in a Biosafety Level 3 (BSL-3) laboratory using BSL-3 practices, and that “site and activity-specific biosafety risk assessments be performed to determine if additional biosafety precautions are warranted based on situational needs” [6]. To facilitate downstream analyses of materials generated from SARS-CoV-2 infection experiments in cell culture there is a need for validated protocols for inactivation of SARS-CoV-2 to facilitate safe transfer of material out of a BSL-3 laboratory.

To date, several protocols that effectively inactivate SARS-CoV-2 in clinical and some research settings have been described, including use of heat to ≥95 °C as a part of diagnostic polymerase chain reaction (quantitative PCR) assays performed in clinical specimens [7,8,9,10], use of heat to 56 °C for 30 min. in serum samples [10], use of alcohol-based hand sanitizers [11], use of ultraviolet (UV) light to disinfect surface contamination [12] use of lysis buffers containing sodium dodecyl sulfate (SDS) or Triton-X100 in cell culture supernatant or nasopharyngeal samples [13,14,15,16], and use of heat to 80 °C, Trizol^®^, detergents and UV energy in stocks of SARS-CoV-2 harvested from infected Vero E6 cells [15]. However, we are not aware of reports validating protocols to inactivate SARS-CoV-2 in differentiated organotypic primary airway epithelial cell (AEC) cultures. We sought to validate a set of BSL-3 SARS-CoV-2 inactivation protocols specific to common cellular and molecular research methods to support isolation of protein from cell layers, harvest of cell culture supernatant for subsequent analyses of secreted constituents, isolation of RNA for use in quantitative PCR assays, and fixation of cells for immunofluorescence confocal microscopy, from differentiated organotypic primary AECs cultures infected with live SARS-CoV-2 ex vivo.

## 2. Methods

### 2.1. Severe Acute Respiratory Syndrome Coronavirus 2 (SARS-CoV-2) Propagation

Vero African green monkey kidney epithelial E6 cells (CRL-1586) were obtained from ATCC^®^ and SARS-CoV-2 isolate USA-WA1/2020 was obtained from National Institutes of Health/National Institute of Allergy and Infectious Diseases supported BEI Resources^®^. All work involving live SARS-CoV-2 was conducted in the Seattle Children’s Research Institute (SCRI) Biosafety Lab Level 3 (BSL-3) and was approved by the Seattle Children’s Institutional Biosafety Committee (IBC). Vero E6 cells were seeded onto T182 flasks (Corning^®^) and grown until they were approximately 75% confluent resulting in a cell count of 7 × 10^6^ per flask. Flasks were then inoculated in a 5% CO_2_ incubator at 37 °C with SARS-CoV-2 in viral growth medium (Dulbecco’s Modified Eagle Medium supplemented with 2% heat-inactivated fetal bovine serum, 2 mM L-glutamine, 100 units of Penicillin and 100 ug/mL Streptomycin) at a multiplicity of infection (MOI) of 0.5 for 2 h using viral stock with a titer of 3.5 × 10^6^ plaque-forming units (PFU)/mL, rocking flasks gently every 15 min. Medium with viral inoculum was then gently removed, flasks were rinsed with phosphate buffered saline (PBS), then 25 mL of viral growth medium was added to flasks. Cells were incubated in a 5% CO_2_ incubator at 37 °C for approximately 48 h when 75% of cells demonstrated cytopathic effect (CPE). Cell lysate was collected from flasks by gentle scrapping and pipetting, then centrifuged at 3000 RPM for 10 min at 4 °C. Supernatant was then aliquoted into sterile vials and stored at −80 °C. 

### 2.2. Determination of Viral Titer

SARS-CoV-2 viral titer was determined in confluent Vero E6 cell cultures in 48 well plates using a 50% tissue culture infectious dose assay (TCID50) [12,17]. Serial dilutions of virus samples from NEAT through 10^−7^ were added to confluent Vero E6 cultures, incubated in a 5% CO_2_ incubator at 37 °C for 72 h then examined for CPE in infected cells, as described by Ksiazek et al. [18]. SARS-CoV-2-induced CPE of infected cells was determined by observing rounding and detachment of cells or complete destruction of cells in the wells inoculated with virus for each dilution. Media was then removed from wells and fixed in 100 uL of 10% formaldehyde for 30 min. Following fixation, formaldehyde was removed and 100 uL of 1% crystal violet (Sigma-Aldrich^®^, St Louis, MO, USA) was added and left on wells for 30 min at room temperature. Crystal violet was then removed, and wells were rinsed twice with tap water then allowed to dry for at least 2 h. Wells were scored as positive for CPE if no, minimal, or patchy crystal violet staining was present, or negative for CPE if wells demonstrated uniform robust crystal violet staining. The titer of SARS-CoV-2 was calculated by the method of Reed and Muench [17].

### 2.3. Assays for Verification of Inactivation Methods

Vero E6 cells were seeded at 1 × 10^5^ cells/well into 48 well plates in Vero expansion media (Eagle’s minimal essential medium supplemented with 2% heat inactivated FBS, 100 units of Penicillin and 100 ug/mL Streptomycin; ATCC^®^). Cells were cultured overnight in a 5% CO_2_ incubator at 37 °C and checked 24 h later to ensure confluency. To test whether a specific processing method inactivated SARS-CoV-2, first media was removed from each Vero cell well and replaced with 100 μL of material that was harvested from SARS-CoV-2-infected differentiated organotypic human airway epithelial cell cultures and inactivated using a proposed method. Material was inoculated NEAT and in 10-fold PBS dilutions from 1 × 10^−1^ through 1 × 10^−7^ onto the surface of confluent Vero E6 cells, with triplicate wells for each concentration of material tested. Material tested was left on the surface of Vero cells for 1 h, rocking the plate gently every 15 min, then removed and replaced with 300 μL of Vero growth medium. For blank control wells, media was replaced with 100 μL of PBS for 1 h, then pipetted off and replaced with 300 μL viral growth medium. Vero cell cultures were then incubated in a 5% CO_2_ incubator at 37 °C for 72 h. Media was then removed from wells and fixed in 100 uL of 10% formaldehyde for 30 min. Following fixation, formaldehyde was removed and 100 uL of 1% crystal violet (Sigma-Aldrich^®^, St Louis, MO, USA) was added and left on wells for 30 min at room temperature. Crystal violet was then removed, and wells were rinsed twice with tap water and wells were allowed to dry for at least 2 h. Wells from each inactivation method were then scored as positive for CPE if any well at a specific dilution demonstrated no, patchy, or minimal crystal violet staining, and scored as negative for infection if all wells at a specific dilution demonstrated uniform robust crystal violet staining. 

### 2.4. Differentiation of Primary Airway Epithelial Cells and Infection with SARS-CoV-2

Using bronchial AECs from three healthy children obtained under study #12490 approved by the Seattle Children’s Institutional Review Board, with investigations carried out following the rules of the Declaration of Helsinki of 1975, AECs were differentiated for 21 days at an air–liquid interface (ALI) on 12-well collagen-coated Corning^®^ plates with permeable transwells in PneumaCult™ ALI media (Stemcell™, Vancouver, BC, Canada) at 37 °C in an atmosphere of 5% CO_2_ as we have previously described, producing an organotypic differentiated epithelial culture with mucociliary morphology [19,20,21,22,23,24]. Differentiated AECs were then infected with SARS-CoV-2 by pipetting 100 μL of viral stock (with a titer of 3.5 × 10^6^ PFU/mL) onto the surface at a MOI of 0.5. Cell lysate, RNA, and supernatant was harvested, and AECs were prepped for immunohistochemistry (IHC), 96 h following infection with SARS-CoV-2. Cell layer material from SARS-CoV-2 infected AECs was scrapped from transwells using a pipet tip, suspended in PBS, then inoculated onto Vero E6 cells for positive control TCID50 assays. Similarly, supernatant from SARS-CoV-2 infected AECs was inoculated onto Vero cells as positive controls. To measure viral replication in AEC cultures we used quantitative PCR, with triplicate assays of harvested RNA from each SARS-CoV-2-infected AEC donor cell line (Genesig^®^ Coronavirus Strain 2019-nCoV Advanced PCR Kit, Primerdesign^®^, Southampton, UK). Concentration of RNA harvested from AECs was determined by Nanodrop^®^.

### 2.5. Heat Inactivation of SARS-CoV-2

Aliquots of SARS-CoV-2 viral stock (3.5 × 10^6^ PFU/mL) were placed in a heat block at 65 °C for 10 min. Viral stocks were allowed to cool to 37 °C before they were used to inoculate Vero E6 cells. 

### 2.6. Isolation of Protein from SARS-CoV-2 Infected Primary Airway Epithelial Cell (AEC) Cultures

To extract protein from the cell layer of SARS-CoV-2-infected AEC cultures, media was first removed from the basolateral chamber of transwells. Next, 100 µL of cold PBS was added to the apical surface of cultures and 1 mL was added to the basolateral chamber of cultures as a wash step. Next, 50 μL of RIPA buffer for protein extraction ready-to-use-solution (Sigma-Aldrich^®^, St Louis, MO, USA, Product No. R0278) containing Triton X100 1% and SDS 0.1% was added to the apical surface of AECs and incubated for 15 min on ice. A pipet tip was then used to gently scratch each apical well in a crosshatch pattern then around the edges of wells to loosen AECs from the transwell membrane. Material was collected, centrifuged at 10,000 rpm at 4 °C for 10 min, then supernatant containing isolated protein was collected for inactivation assays.

### 2.7. Ultraviolet (UV) Light Treatment of Supernatant from SARS-CoV-2 Infected Vero E6 Cells and Primary AEC Cultures

Supernatant from SARS-CoV-2 infected Vero E6 cells or AECs were removed from their respective cultures and placed into 12 well plates, with 1 mL of supernatant in each well. The 12-well plate containing supernatant from Vero cell or AEC cultures was then placed into a VWR^®^ UV Crosslinker 254 nm UV (VWR catalog # 89131-484) set to deliver 200,000 microjoules over a 5-min exposure.

### 2.8. Isolation of RNA from SARS-CoV-2 Infected Primary AEC Cultures

RNA was isolated from SARS-CoV-2 infected AECs using a TRIzol™ Plus RNA Purification Kit (ThermoFisher Cat #: 12183555). 500 uL of TRIzol™ was added to the apical surface of the transwell inserts, transwells were scraped in a crosshatch pattern to loosen the cells, then TRIzol™ was allowed to incubate for 5 min. Material was then added to 100 uL of chloroform (0.2 mL chloroform per 1 mL Trizol™) and incubated for 2–3 min, then centrifuged for 15 min at 12,000× *g* at 4 °C. 600 µL of the upper aqueous phase containing the RNA was transferred to a new tube to which an equal volume of 70% ethanol was added and the material vortexed. Thereafter, RNA was isolated following manufacturer protocols using appropriate TRIzol™ wash buffers and spin cartridges with final sample elution with RNase-free water (TRIzol™ Plus RNA Purification Kit, ThermoFisher^®^, Waltham, MA, USA).

### 2.9. Preparation of AECs for Immunofluorescence Confocal Microscopy

SARS-CoV-2 infected AEC transwell inserts were fixed for IHC by adding 600 uL of 10% formalin to the basolateral chamber and 200 uL of 10% formalin to the apical surface of AEC transwells at room temperature for 30 min. Next, apical and basolateral surfaces of transwell inserts were washed with PBS 3 times, then 1% Triton X-100 in PBS was added to the apical surface of fixed cells for 5 min to increase permeability prior to incubations with primary and secondary antibodies of interest. To test the ability of formalin and Triton-X exposures in this protocol to inactivate SARS-CoV-2, we trypsinized infected AECs then exposed the cells to 10% formalin for 30 min and 1% Triton X-100 for 5 min, then exposed Vero cells to formalin and Triton X-100 treated material. To demonstrate SARS-CoV-2 viral particles are present in inactivated IHC preparations from SARS-CoV-2 infected cultures, AECs were stained with a primary antibody to the SARS-CoV-2 spike protein (1:100; Mouse anti-COVID 19 Spike Receptor Binding Domain Monoclonal Antibody, MyBioSource, San Diego, CA, USA) and a secondary anti-mouse antibody (1:1000; Alexa Fluor 488 donkey anti-mouse IgG, ThermoFisher, Waltham, MA, USA).

## 3. Results

First, confluent Vero E6 cells grown on 48 well plates were inoculated with 100 μL of NEAT SARS-CoV-2 viral stock with a titer of 3.5 × 10^6^ PFU/mL, and in 10-fold PBS dilutions from 1 × 10^−1^ through 1 × 10^−7^ for 1 h. Virus was removed and cultures were incubated at 37 °C in an atmosphere of 5% CO_2_ for 72 h, then cultures were fixed with formaldehyde and stained with 1% crystal violet. All wells from NEAT through 1 × 10^−7^ dilution were positive for evidence of SARS-CoV-2 infection (Figure 1). When Vero E6 cells were inoculated with SARS-CoV-2 viral stock which had been heated to 65 °C for 10 min we observed normal crystal violet staining in all wells consistent with no infection in wells inoculated with NEAT virus or any dilution through 1 × 10^−7^ (Figure 1). 

Primary AEC cultures using cells from three pediatric donors were infected with SARS-CoV-2 at a MOI of 0.5. Confocal microscopy of differentiated AECs with a fluorescent antibody to the SARS-CoV-2 spike protein demonstrated diffuse infection of the apical layer of AEC cultures (Figure 2). The AEC cell layer harvested from SARS-CoV-2 infected cultures was confirmed to be highly infectious in Vero cell TCID50 assays (Figure 2E). Surprisingly, supernatant from SARS-CoV-2 infected AECs did not demonstrate evidence of infection in Vero E6 cells (Figure 2E). RNA was harvested from uninfected AECs, SARS-CoV-2-infected AEC cultures, and AECs exposed to UV-inactivated SARS-CoV-2, 96 h following viral inoculation to measure viral copy number by qPCR as a measure of SARS-CoV-2 replication. Differentiated primary human AEC cultures are inherently heterogenous between human donors and contain variable amounts of mucus secreted onto the apical surface, thus limiting the utility of apical washes or harvested cell layer material from infected cultures as objective outcome measures between individuals or between experimental conditions to quantify infectious virus. Therefore, we assessed viral copy number by qPCR, which can be normalized to the RNA content of a particular primary AEC culture, providing a more objective and consistent viral replication outcome measure that can be used across a large number of primary AEC cultures and experimental conditions. Between the 3 donor primary AEC cultures, raw viral copy number ranged from 2.3 × 10^5^ to 3.2 × 10^6^ (median 2.8 × 10^6^; Figure 3A), and when adjusted for RNA concentration ranged from 2.1 × 10^3^ to 1.2 × 10^4^ (median 1.1 × 10^4^; Figure 3B.).

Confluent Vero E6 cells were first inoculated with 100 μL of NEAT protein which had been isolated from uninfected primary human AECs using 1% Triton X100 and 0.1% SDS (in RIPA buffer), and in 10-fold PBS dilutions of isolated protein from 1 × 10^−1^ through 1 × 10^−7^, to determine if there was any toxicity to Vero cells from residual chemicals from the protein isolation protocol. We observed toxicity to Vero cells in the NEAT and the 1 × 10^−1^ dilution with complete elimination of crystal violet staining secondary to cell lysis from the RIPA buffer containing 0.1% SDS (Figure 4). Next, to verify inactivation of SARS-CoV-2 in protein isolated from infected AECs, we inoculated Vero cells with protein isolated from AECs that had been infected with SARS-CoV-2 at an MOI of 0.5 for 96 h, using NEAT protein and 10-fold PBS dilutions of protein from 1 × 10^−1^ through 1 × 10^−7^. In the 1 × 10^−1^ dilution we observed the same pattern of complete elimination of crystal violet staining secondary to cell lysis from SDS as was noted when Vero cells were exposed to protein isolated from uninfected AECs. We observed normal crystal violet staining consistent with no viral infection for Vero cells exposed to protein isolated from SARS-CoV-2 infected AECs that was diluted 1 × 10^−2^ through 1 × 10^−7^ (Figure 4 and Figure 5). A limitation of our approach was that we did not utilize methods to remove Triton X and SDS from SARS-CoV-2 infected and Triton X/SDS treated cell layer to definitively prove that virus inactivation occurred at the 1 × 10^−1^ dilution. However, Triton X and SDS have been shown to inactivate SARS-CoV-2 in other recent publications [15,16]. Furthermore, when we performed quantitative PCR on material isolated using our protein extraction protocol we observed a median SARS-CoV-2 copy number of 4.4 × 10^4^ (Figure 4C) which should have resulted in cytopathic effect in Vero cells at dilutions significantly greater than the 1 × 10^−1^ dilution if the protein isolate samples had not been inactivated.

Confluent Vero E6 cells were exposed to uninfected AECs that had been treated with 10% formalin for 30 min and 1% Triton X-100 for 5 min, at NEAT and 10-fold PBS dilutions from 1 × 10^−1^ through 1 × 10^−7^. The NEAT concentration was noted to be toxic to cells (Figure 3). Next, to verify inactivation of SARS-CoV-2 in AECs treated with 10% formalin and 1% Triton X-100, we inoculated Vero cells with AECs that had been infected with SARS-CoV-2 at an MOI of 0.5 for 96 h then treated with 10% formalin and 1% Triton X-100 using 10-fold PBS dilutions of treated AECs from 1 × 10^−1^ through 1 × 10^−7^. We observed normal crystal violet staining consistent with no viral infection for Vero cells exposed to SARS-CoV-2 infected AECs treated with formalin and Triton X-100 that was diluted 1 × 10^−1^ through 1 × 10^−7^ (Figure 4 and Figure 5).

To verify the inactivation of SARS-CoV-2 in supernatant from infected AEC cultures, supernatant was exposed to UV light in a UV Crosslinker that delivered 200,000 microjoules over a 5-min exposure. UV-exposed AEC supernatant was then used to inoculate Vero E6 cells, using NEAT supernatant and supernatant diluted in PBS from 1 × 10^−1^ through 1 × 10^−7^. We observed normal crystal violet staining consistent with no viral infection for Vero cells exposed to NEAT and diluted UV-exposed supernatant, confirming that UV light inactivated supernatant form SARS-CoV-2 infected AEC cultures (Figure 4). 

To verify the inactivation of SARS-CoV-2 in RNA isolated using TRIzol™ from infected AEC cultures RNA isolated from infected AECs was used to inoculate Vero cells using NEAT RNA and RNA diluted in PBS from 1 × 10^−1^ through 1 × 10^−7^. We observed normal crystal violet staining consistent with no viral infection for Vero cells exposed to NEAT and diluted RNA, confirming that isolation of RNA using TRIzol™ inactivated SARS-CoV-2 (Figure 4 and Figure 5).

Finally, to determine if our protocol to prepare AEC cultures for IHC inactivates SARS-CoV-2, we first trypsinized SARS-CoV-2 infected AEC cultures then exposed the cells to 10% formalin for 30 min and 1% Triton X-100 for 5 min. Next, we exposed Vero cells to formalin and Triton X-100 treated material NEAT and diluted in PBS from 1 × 10^−1^ through 1 × 10^−7^. We observed normal crystal violet staining consistent with no viral infection for Vero cells exposed to formalin and Triton X-100 treated material, confirming that our AEC IHC preparation protocol inactivated SARS-CoV-2 (Figure 4 and Figure 5).

## 4. Discussion

There is an urgent need for research on SARS-CoV-2, the virus that causes COVID-19. It is of particular importance to improve our understanding of mechanisms that underlie the heterogeneity of disease severity with SARS-CoV-2 infection between individuals and explain why children are more resistant to developing severe COVID-19 than adults. Such knowledge will be critical in developing novel therapeutic interventions to treat and prevent SARS-CoV2 infection and COVID19 disease. Given that the airway epithelium is the initial site of infection, study of organotypic primary human AEC cultures infected with SARS-CoV-2 will be crucial to improving our understanding of heterogeneity between individuals in how the virus gains entry to AECs as well as innate immune responses to the virus by AECs. To initiate such investigations we conducted a series of experiments which validate BSL-3 SARS-CoV-2 inactivation protocols to support the isolation of the protein, harvest of cell culture supernatant for subsequent analyses of secreted constituents, isolation of RNA for use in quantitative PCR assays, and fixation of cells for IHC, from SARS-CoV-2-infected organotypic primary AEC cultures. Interestingly, inoculation of Vero cells with supernatant from SARS-CoV-2 infected AECs did not produce evidence of infection, suggesting that SARS-CoV-2 is not easily spread basolaterally in these organotypic AEC cultures. However, despite this observation we recommend that supernatant from SARS-CoV-2 infected AECs should still be inactivated with UV light given that our data cannot definitively rule out the possibility of some viral shedding into supernatant in some differentiated AEC cultures. We and others have shown that UV light effectively inactivates SARS-CoV-2 harvested from infected Vero E6 cells [12,15]. Together, these protocols allow for biomaterials from SARS-CoV-2 infection experiments in differentiated primary AECs to be safely transferred out of a BSL-3 laboratory for further analyses or use in specialized assays that are often not practical in a BSL-3 facility. We believe that these protocols will be useful to other laboratories conducting live SARS-CoV-2 work in primary AECs and contribute to efforts to improve understanding of the heterogeneity of airway epithelial host factors that determine SARS-CoV-2 replication and innate immune responses.

## Figures and Tables

**Figure 1 mps-04-00007-f001:**
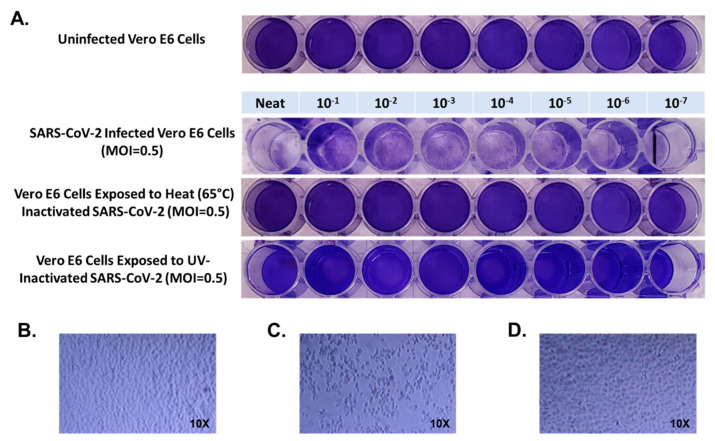
Each experimental condition, and dilution within each condition, were completed in triplicate. Results were consistent within each dilution, therefore a single well per dilution is presented for simplicity. Panel (**A**) presents crystal violet stained uninfected Vero E6 cells, Vero E6 cells infected with severe acute respiratory syndrome coronavirus 2 (SARS-CoV-2) at multiplicity of infection of multiplicity of infection (MOI) = 0.5, Vero E6 cells exposed to SARS-CoV-2 that had been exposed to heat (65 °C) for 10 min, and Vero E6 cells exposed to SARS-CoV-2 that had been treated with UV light (200,000 microjoules over a 5-min). Panel (**B**) presents light microscopy images of uninfected Vero E6 cells, Panel (**C**) presents light microscopy images of Vero E6 cells infected with SARS-CoV-2, and Panel (**D**) presents light microscopy images of Vero E6 cells exposed to heat-inactivated SARS-CoV-2.

**Figure 2 mps-04-00007-f002:**
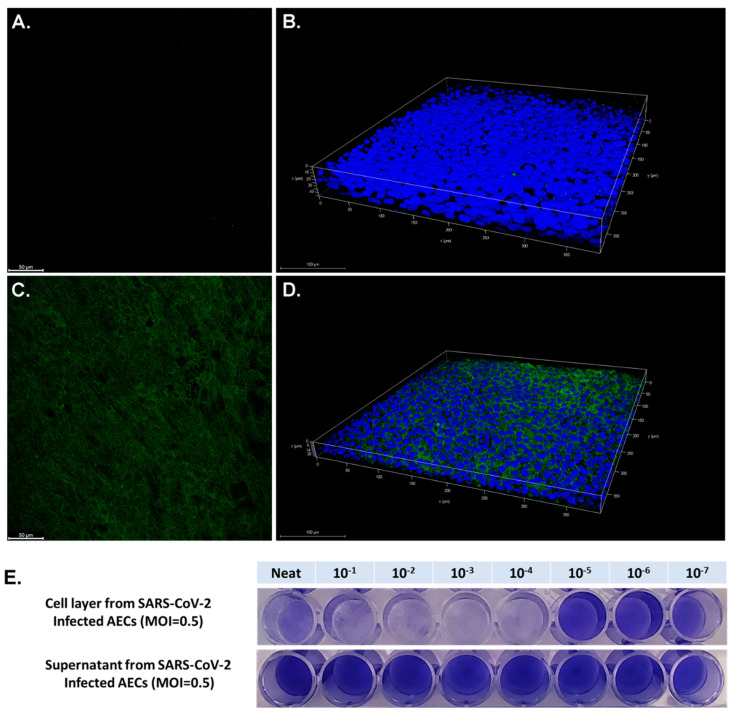
Confocal images of uninfected differentiated airway epithelial cells (AECs) and AECs following infection with SARS-CoV-2 at a MOI of 0.5 for 96 h labeled for spike protein (green) and 4′,6-diamidino-2-phenylindole (DAPI) (blue). Panel (**A**) presents an image obtained from the apical portion of the differentiated uninfected AEC layer. Panel (**B**) presents a 3-dimensional reconstruction of the uninfected AEC layer. Panel (**C**) presents an image obtained from the apical portion of a SARS-CoV-2-infected AEC layer. Panel (**D**) presents a 3-dimensional reconstruction of the SARS-CoV-2-infected AEC layer. Panel (**E**) presents crystal violet stained Vero cells exposed to cell layer harvested from positive control SARS-CoV-2-infected AECs (Neat concentration represents all cell layer material scrapped from an AEC transwell suspended in 100 µL of phosphate buffered saline) and supernatant from SARS-CoV-2 infected AECs. Positive control SARS-CoV-2 infected AEC experiments were completed using 3 separate human primary AEC lines (3 human donors) with consistent results from all three AEC lines.

**Figure 3 mps-04-00007-f003:**
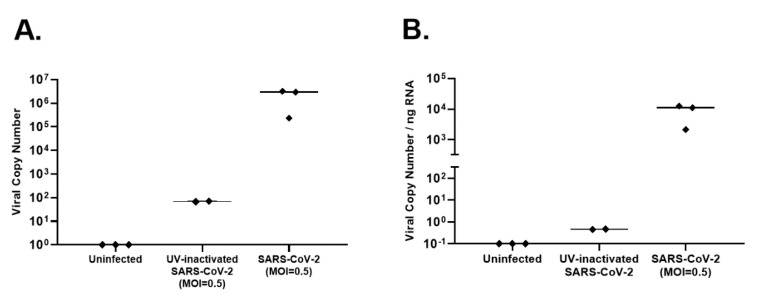
Viral replication in SARS-CoV-2-infected organotypic primary human airway epithelial cells (AEC). Viral copy number was measured by performing quantitative polymerase chain reaction (qPCR) using SARS-CoV-2 inactivated RNA isolated 96 h following inoculation of AECs with SARS-CoV-2 at a MOI of 0.5 or UV-inactivated SARS-CoV-2. Panel (**A**). presents unadjusted copy number and Panel (**B**). presents copy number adjusted for RNA concentration.

**Figure 4 mps-04-00007-f004:**
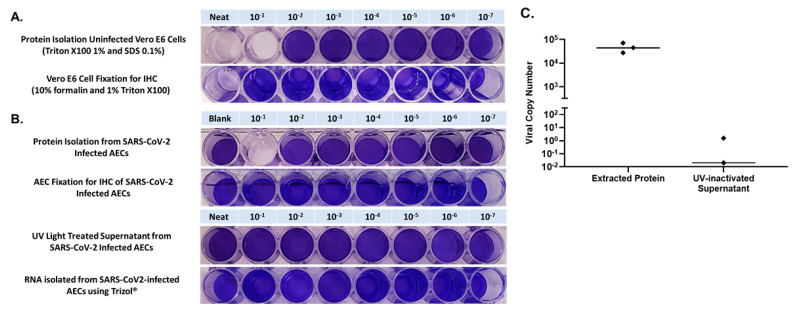
Each experimental condition, and dilution within each condition, were completed in triplicate using 3 separate human primary AEC lines (3 human donors). Results were consistent within each dilution, therefore a single dilution series from one of the 3 primary AEC lines is presented for simplicity. Panel (**A**) presents crystal violet stained uninfected Vero E6 cells exposed to harvested differentiated airway epithelial cell (AEC) cultures treated with Triton X100 1% and sodium dodecyl sulfate (SDS) 0.1% (protein isolation method), or 10% formalin and Triton X100 1% (immunohistochemistry fixation method). AEC material treated with SDS 0.1% resulted in lysis of Vero cells at NEAT and 10^−1^ dilution, and exposure of Vero cells to NEAT material from AECs fixed with 10% formalin and Triton X100 1% also resulted in cell death. Panel (**B**) presents crystal violet stained Vero cells exposed to protein isolated from SARS-CoV-2-infected AECs, material from SARS-CoV-2-infected AECs fixed for immunohistochemistry (IHC), ultraviolet (UV)-light treated supernatant from SARS-CoV-2-infected AECs, and RNA isolated with Trizol^®^ from SARS-CoV-2-infected AECs. The 10^−1^ dilution of exposure to protein isolated from SARS-CoV-2-infected AECs was not scored as positive given that the same dilution of uninfected AECs treated with SDS 0.1% was toxic to Vero cells. Panel (**C**) presents SARS-CoV-2 viral copy number by quantitative PCR of inactivated material isolated for protein analysis using Radioimmunoprecipitation assay buffer (containing Triton X100 1% and SDS 0.1%) and of UV-inactivated supernatant. Using extracted protein material, RNA was secondarily isolated using Trizol^®^ before conducting PCR. Similarly, following UV-inactivation, RNA was isolated from the supernatant using Trizol^®^ before conducting PCR.

**Figure 5 mps-04-00007-f005:**
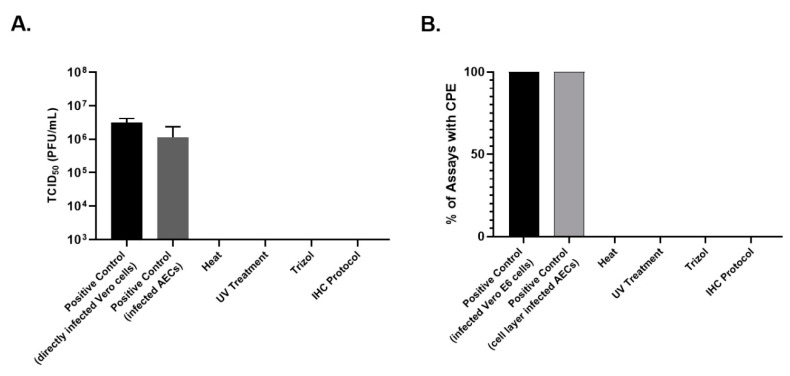
Summary of viral titers from 50% tissue culture infectious dose (TCID_50_) assays Panel (**A**) and observed cytopathic effect (CPE) Panel (**B**) in SARS-CoV-2 infected Vero E6 cells, Vero cells exposed to SARS-CoV-2 infected AEC cell layer, SARS-CoV-2 exposed to heat (65 °C) for 10 min, SARS-CoV-2 treated with UV light, RNA harvested with Trizol^®^ from SARS-CoV-2 infected AECs, and SARS-CoV-2 infected AECs fixed for immunohistochemistry (IHC) from SARS-CoV-2-infected AECs. Given that the protein isolation protocol tested (Triton X and SDS) resulted in direct toxicity with complete CPE at 10^−1^ dilution without SARS-CoV2 (see Figure 4A), this inactivation method is not included in this figure.

## Data Availability

Data sharing not applicable.

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
