# Peer review of "Inactivation of Material from SARS-CoV-2-Infected Primary Airway Epithelial Cell Cultures"

_mps, 2021, doi:10.3390/mps4010007_

Round 1
Reviewer 1 Report
In their manuscript entitled "Inactivation of material from SARS-CoV-2 infected primary airway epithelial cell cultures," Barrow et al. describe methods that are effective in inactivating SARS-CoV-2 in infected airway epithelium cultures by heat, 1% triton x-100/0.1% SDS, 10% formalin/1% triton x-100, and trial extracted RNA. Overall the study is straightforward, however, there are a few points I think the authors should address that would strengthen the manuscript.
Major comments:
- In figure 3 A/B, the authors mention within the text that they identify clear toxicity with neat and 10-fold diluted RIPA buffer samples, but there is no further discussion about this issue. How can the authors conclude that all virus has been inactivated when the least dilute samples result in cytotoxicity, and thus infectious virus cannot be measured in these samples? Given that methods exist for removing detergents from samples, and other SARS-CoV-2 inactivation manuscripts ave employed these methods successfully (PMIDs 32798217, 32839250), the authors should at a minimum discuss this point further and address this as a limitation of the study.
- In figure 3 A/B there is no positive control for the infected Vero cells or AECs shown in this figure. A positive control for both should be included. This could be accomplished by titering the infected cell supernatants or by scraping off the cells and titrating to show the viral load that was inactivated. Figure 1 depicts a positive control for the heat treatment, and I have to assume this is what is being referenced in Figure 4. Given that the experimental design for Figure 3 is different from Figure 1 this should be addressed and new controls added.
- The authors should include the volume of basolateral media that was inactivated by UV in the methods. It's well known that the efficacy of UV light inactivation of viruses is dependent on the volume being inactivated.
- When TCID50 is the preferred method of virus quantification for the majority of this study, why was qPCR used in Figure 2 as opposed to measuring infectious virus being released on the apical surface of the infected AEC trans-wells? This should be addressed in the text.
Minor comments:
1. There are now several published manuscripts regarding the inactivation of SARS-CoV-2. The authors should consider additional references in their introduction to more accurately reflect the number of studies that have tested the inactivation methods described in the introduction.
Author Response
Reviewer #1
In their manuscript entitled "Inactivation of material from SARS-CoV-2 infected primary airway epithelial cell cultures," Barrow et al. describe methods that are effective in inactivating SARS-CoV-2 in infected airway epithelium cultures by heat, 1% triton x-100/0.1% SDS, 10% formalin/1% triton x-100, and trial extracted RNA. Overall the study is straightforward, however, there are a few points I think the authors should address that would strengthen the manuscript.
Major comments:
- In figure 3 A/B, the authors mention within the text that they identify clear toxicity with neat and 10-fold diluted RIPA buffer samples, but there is no further discussion about this issue. How can the authors conclude that all virus has been inactivated when the least dilute samples result in cytotoxicity, and thus infectious virus cannot be measured in these samples? Given that methods exist for removing detergents from samples, and other SARS-CoV-2 inactivation manuscripts have employed these methods successfully (PMIDs 32798217, 32839250), the authors should at a minimum discuss this point further and address this as a limitation of the study.
Response: Thank you for this critique. A limitation of our methods was that we did not utilize methods to remove Triton X and SDS from SARS-CoV-2 infected and Triton X/SDS treated cell layer to prove that virus inactivation occurred in our TCID50 assay at 10-1dilution. However, as noted by this reviewer 1% Triton X and SDS have been shown to inactivate SARS-CoV-2 in other recent papers (PMIDs 32798217, 32839250). We have added this limitation to the revised manuscript.
- In figure 3 A/B there is no positive control for the infected Vero cells or AECs shown in this figure. A positive control for both should be included. This could be accomplished by titering the infected cell supernatants or by scraping off the cells and titrating to show the viral load that was inactivated. Figure 1 depicts a positive control for the heat treatment, and I have to assume this is what is being referenced in Figure 4. Given that the experimental design for Figure 3 is different from Figure 1 this should be addressed and new controls added.
Response: There is a positive control for infected Vero cells in Figure 1 (panel A, row 2), however, we agree that the lack of positive control SARS-CoV-2 infected AECs was a limitation in our original submission. We have now added a new Figure 2 that includes TCID50 assays for SARS-CoV-2 infected AECs from both the AEC cell layer and AEC supernatant (panel E). We have also added confocal microscopy images of SARS-CoV-2 infected AECs with fluorescent labeling of the SARS-CoV-2 spike protein to this new Figure 2 (panels C and D). Interestingly, TCID50 assays of supernatant from SARS-CoV-2 infected AECs (three different AEC primary lines) did not reveal any cytopathic effect, suggesting that SARS-CoV-2 is not easily spread basolaterally in these organotypic AEC layers. We have added sentences to the Discussion acknowledging this finding. However, despite these observations we recommend that supernatant from SARS-CoV-2 infected AECs should still be inactivated with UV light given that our data cannot definitively rule out some viral shedding into supernatant in all differentiated AEC cultures, and we (and others) have shown that UV light effectively inactivates virus harvested from infected Vero E6 cells (we have added to Figure 1 a row demonstrating this). Old Figures 2-onward have been renumbered.
- The authors should include the volume of basolateral media that was inactivated by UV in the methods. It's well known that the efficacy of UV light inactivation of viruses is dependent on the volume being inactivated.
Response: We have added the volume of supernatant from Vero cells or AECs inactivated by UV light (1mL in each well).
- When TCID50 is the preferred method of virus quantification for the majority of this study, why was qPCR used in Figure 2 as opposed to measuring infectious virus being released on the apical surface of the infected AEC trans-wells? This should be addressed in the text.
Response: We have added the following text: “Differentiated primary human AEC cultures are inherently heterogenous between human donors and contain variable amount of mucus secreted onto the apical surface, thus limiting the utility of apical washes or harvested cell layer material from infected cultures as objective outcome measures between individuals or between experimental conditions quantify infectious virus. Therefore, we assessed viral copy number by qPCR, which can be normalized to the RNA content of a particular primary AEC culture, providing a more objective and consistent viral replication outcome measure that can be used across a large number of primary AEC cultures and experimental conditions”.
We have also modified Figure 3 (old Figure 2) to include viral copy number from AEC cultures exposed to UV-inactivated SARS-CoV-2 for context.
Minor comments:
- There are now several published manuscripts regarding the inactivation of SARS-CoV-2. The authors should consider additional references in their introduction to more accurately reflect the number of studies that have tested the inactivation methods described in the introduction.
Response: We have added relevant new published manuscripts as references in our Introduction [PMIDs 32798217, 32839250, 32838945].
Reviewer 2 Report
This manuscript would be of interest to those researching SARS-CoV-2. Several inactivation methods are already known and published, this manuscript aims to add to that existing knowledge with the addition of samples from primary AEC cultures.
The technical approaches to answering the question of efficacy of different methods of inactivation are sound however, there are some obvious omissions that would need corrected prior to publication.
Most importantly, the plaque assays of paired samples which have not undergone inactivation should be shown. Namely, in figure 3 plaque assays from UV treated supernatant (is it apical or basolateral?) from AECs are shown, but the corresponding plaque assays from un-treated samples are not. The authors need to establish that there is a productive SARS-CoV-2 infection with virus release into the supernatant before they can state their activation protocol is effective.
Similarly to the point above, Figure 2 shows the donor variation in viral copy number of AECs from RNA extraction of the cultures. Plaque assays of the supernatant (preferably apical and basolateral) should be carried out to show released virus. This would also be of interest to show if there is virus released into the basolateral medium, which is still debated.
The authors state, in the figure legend of Figure 3, that each experimental condition was completed in triplicate. For confidence in the results it should be known if these were the same samples from AECs inactivated and plaque assays were carried out in triplicate. Or was the whole experiment, including infection, samples and inactivation carried out 3 times and each was plaqued? Do these 3 experiments correspond to the 3 donors?
It would be beneficial if the authors were able to quantify their plaque assay results. Although I understand that plaques for SARS-CoV-2 can be difficult to count (especially without the furin cleavage site mutation).
The IHC inactivation protocol seems sound. It would be good to see the IHC images and show that detection of viral proteins is still possible following this inactivation and permeabilisation.
In Fig 4B the author shows 0% of assays with CPE for the inactivation methods. However, some inactivation methods were shown to be toxic in higher concentrations. It is not accurate to say there is 0% CPE if the cultures show damage due to cytotoxicity. This should be taken into consideration.
Minor points: revise sentence structure in lines 212-217, it is unclear. The punctuation after 'hrs' is not needed. Line 215: word 'by' not needed.
This reviewer has always written dilutions as i.e 10-7 (as opposed to 107 in the manuscript).
Author Response
Reviewer #2
This manuscript would be of interest to those researching SARS-CoV-2. Several inactivation methods are already known and published, this manuscript aims to add to that existing knowledge with the addition of samples from primary AEC cultures.
The technical approaches to answering the question of efficacy of different methods of inactivation are sound however, there are some obvious omissions that would need corrected prior to publication.
Most importantly, the plaque assays of paired samples which have not undergone inactivation should be shown. Namely, in figure 3 plaque assays from UV treated supernatant (is it apical or basolateral?) from AECs are shown, but the corresponding plaque assays from un-treated samples are not. The authors need to establish that there is a productive SARS-CoV-2 infection with virus release into the supernatant before they can state their activation protocol is effective.
Response: We appreciate this critique which is similar to Reviewer #1, question 2. We agree that the lack of positive control SARS-CoV-2 infected AECs was a limitation in our original submission. We have now added a new Figure 2 that includes TCID50 assays for SARS-CoV-2 infected AECs from both the AEC cell layer and AEC supernatant (panel E). We have also added confocal microscopy images of SARS-CoV-2 infected AECs with fluorescent labeling of the SARS-CoV-2 spike protein to this new Figure 2 (panels C and D). Interestingly, TCID50 assays of supernatant from SARS-CoV-2 infected AECs (three different AEC primary lines) did not reveal any cytopathic effect, suggesting that SARS-CoV-2 is not easily spread basolaterally in these organotypic AEC layers. We have added sentences to the Discussion acknowledging this finding. However, despite these observations we recommend that supernatant from SARS-CoV-2 infected AECs should still be inactivated with UV light given that our data cannot definitively rule out some viral shedding into supernatant in all differentiated AEC cultures, and we (and others) have shown that UV light effectively inactivates virus harvested from infected Vero E6 cells (we have added to Figure 1 a row demonstrating this). Old Figures 2-onward have been renumbered.
Similarly to the point above, Figure 2 shows the donor variation in viral copy number of AECs from RNA extraction of the cultures. Plaque assays of the supernatant (preferably apical and basolateral) should be carried out to show released virus. This would also be of interest to show if there is virus released into the basolateral medium, which is still debated.
Response: We have added positive control TCID50 assays for both cell layer and supernatant from SARS-CoV-2 infected AECs. Interestingly, TCID50 assays of supernatant from SARS-CoV-2 infected AECs (three different AEC primary lines) did not reveal any cytopathic effect, suggesting that SARS-CoV-2 is not easily spread basolaterally in these organotypic AEC layers. We have added sentences to the Discussion acknowledging this finding. However, despite these observations we recommend that supernatant from SARS-CoV-2 infected AECs should still be inactivated with UV light given that our data cannot definitively rule out some viral shedding into supernatant in all differentiated AEC cultures, and we (and others) have shown that UV light effectively inactivates virus harvested from infected Vero E6 cells (we have added to Figure 1 a row demonstrating this). Old Figures 2-onward have been renumbered.
The authors state, in the figure legend of Figure 3, that each experimental condition was completed in triplicate. For confidence in the results it should be known if these were the same samples from AECs inactivated and plaque assays were carried out in triplicate. Or was the whole experiment, including infection, samples and inactivation carried out 3 times and each was plaqued? Do these 3 experiments correspond to the 3 donors?
Response: Experiments were completed in triplicate using 3 separate human primary AEC lines (3 human donors). This has been clarified in the Figure legend.
It would be beneficial if the authors were able to quantify their plaque assay results. Although I understand that plaques for SARS-CoV-2 can be difficult to count (especially without the furin cleavage site mutation).
Response: Th inactivation work in this manuscript was completed using a TCID50 approach as opposed to plaque assays. Estimates of viral titer can be performed using the TCID50 approach (using the method of Reed-Muench). We believe that estimates of SARS-CoV-2 infectious particles from TCID50 assays is only relevant to Figure 1 for the SARS-CoV-2 infected Vero cells and in our new Figure 2 (for the positive control cell layer from the SARS-CoV-2 infected AECs). Quantification of viral infectious particles from both the SARS-CoV-2 infected Vero cells and AECs (positive controls) is also now included in new Figure 4A (revised from old Figure 3).
The IHC inactivation protocol seems sound. It would be good to see the IHC images and show that detection of viral proteins is still possible following this inactivation and permeabilisation.
Response: In newly added Figure 2 we now include IHC images from SARS-CoV-2 infected differentiated AECs demonstrating detection of SARS-CoV-2 with fluorescent labeling of the spike protein.
In Fig 4B the author shows 0% of assays with CPE for the inactivation methods. However, some inactivation methods were shown to be toxic in higher concentrations. It is not accurate to say there is 0% CPE if the cultures show damage due to cytotoxicity. This should be taken into consideration.
Response: Thank you. We have removed Protein Isolation from Figure 5 (old Figure 4).
Minor points: revise sentence structure in lines 212-217, it is unclear. The punctuation after 'hrs' is not needed. Line 215: word 'by' not needed.
Response: Thank you, the sentence structure has been improved and “hrs” has been replaced with hours.
This reviewer has always written dilutions as i.e 10-7 (as opposed to 107 in the manuscript).
Response: Thank you, these have been corrected throughout the manuscript.
Reviewer 3 Report
The inactivation is COVID19 for subsequent use is of value. As also mentioned in the article, it would be important to support subsequent isolation of proteins, isolation of RNA for use in quantitative PCR, etc are important.
However, contrary to this, the results of this paper do not show isolation of proteins or RNA from directly inactivated samples. Rather it is from Vero cells infected with inactivated samples measured by CPE.
If this was not misunderstood, would suggest that the paper touch on the use of such inactivation as most purpose to inactivate samples for for immediate protein or RNA extraction.
Perhaps the authors can approach this as in general inactivation (decontamination) purpose or like to get proteins without active virions or down that direction without trying to mention the purpose for direct downstream work.
Otherwise, will need to show protein/RNA extraction of infected cells that were then inactivated.
Author Response
The inactivation of COVID19 for subsequent use is of value. As also mentioned in the article, it would be important to support subsequent isolation of proteins, isolation of RNA for use in quantitative PCR, etc are important.
However, contrary to this, the results of this paper do not show isolation of proteins or RNA from directly inactivated samples. Rather it is from Vero cells infected with inactivated samples measured by CPE. If this was not misunderstood, would suggest that the paper touch on the use of such inactivation as most purpose to inactivate samples for for immediate protein or RNA extraction.
Response: Thank you for this comment. Actually, with the exception of data in Figure 1 (and associated text) which does present results from infected Vero E6 cells, the rest of the data in this manuscript (Figures 2-5 and associated text) include data from SARS-CoV-2 infected primary airway epithelial cells (AECs) wherein RNA, protein, supernatant, and material for IHC are in fact isolated from infected AEC cultures, then that harvested material from SARS-CoV-2-infected AECs is then tested for inactivation by adding to Vero cells.
Perhaps the authors can approach this as in general inactivation (decontamination) purpose or like to get proteins without active virions or down that direction without trying to mention the purpose for direct downstream work. Otherwise, will need to show protein/RNA extraction of infected cells that were then inactivated.
Response: As noted above, Figure 2 (panel E), Figure 3, Figure 4 (panel B), and Figure 5 all include data using protein, RNA, and/or supernatant extracted from SARS-CoV-2 infected AECs that was then inactivated, with inactivation verified by inoculating this material from infected AECs on Vero cells.
Round 2
Reviewer 1 Report
My concerns have been addressed by the revisions and the manuscript now seems suitable for publication.
Author Response
Thank you, we appreciate your response to our revisions and your contributions to an improved manuscript.
Reviewer 3 Report
Given that the paper is now likely to appear only end Dec early Jan 2021, good to consider updating the latest figures of the 1st sentence of introduction so it feels updated.
Paper would benefit from demonstrating is that for RNA and Protein isolation from the AECs had consistent % of viral RNA over total RNA, and same for protein, or some evidence of detection to strengthen, but this is probably a big ask at this point, but can be considered for future. Because the quantification of the proteins and UV could be used to support the effectiveness of inactivation better.
Author Response
Reviewer #3
Given that the paper is now likely to appear only end Dec early Jan 2021, good to consider updating the latest figures of the 1st sentence of introduction so it feels updated.
Response: We appreciate this critique. Worldwide infection and mortality statistics have been updated.
Paper would benefit from demonstrating is that for RNA and Protein isolation from the AECs had consistent % of viral RNA over total RNA, and same for protein, or some evidence of detection to strengthen, but this is probably a big ask at this point, but can be considered for future. Because the quantification of the proteins and UV could be used to support the effectiveness of inactivation better.
The inactivation is COVID19 for subsequent use is of value. As also mentioned in the article, it would be important to support subsequent isolation of proteins, isolation of RNA for use in quantitative PCR, etc are important.
Response: Thank you for this critique. Although we agree that this would be interesting, at the current time it is not feasible for us to estimate the fraction of viral vs. mammalian RNA in our samples, and we believe this question is beyond the scope of this paper which was intended to further validate SARS-CoV-2 inactivation protocols to allow for safe removal of deactivated samples from our differentiated airway epithelial model system from a BSL-3 facility to allow for further analysis. However, to address the request by Reviewer #3 for evidence of detection of viral RNA in inactivated protein and UV-treated supernatant samples we have made the following minor modifications/additions to our manuscript:
- We have clarified in the legend of Figure 3 to state that viral copy number was determined “by performing quantitative polymerase chain reaction (PCR) using SARS-CoV-2 inactivated RNA” isolated 96 hours following inoculation of AECs with SARS-CoV-2.
- We have added a new panel C to Figure 4 that presents SARS-CoV-2 viral copy number by quantitative PCR of inactivated material isolated and inactivated for protein analysis using RIPA buffer (containing Triton X100 1% and SDS 0.1%), and of UV-inactivated supernatant. Using extracted protein material, RNA was secondarily isolated from the material using Trizol® before conducting PCR. Similarly, following UV-inactivation, RNA was isolated from the supernatant using Trizol® before conducting PCR. For the material isolated and inactivated for protein analysis this additional data provides evidence of viral RNA “detection” with a high copy number as requested by Reviewer #3. We have also added the following text to the body of the manuscript (lines 283-287) to accompany this additional data: “Furthermore, when we performed quantitative PCR on material isolated using our protein extraction protocol we observed a median SARS-CoV-2 copy number of 4.4 x 104 (Fig. 4, panel C) which should have resulted in cytopathic effect in Vero cells at dilutions significantly greater than the 1x10-1 dilution if the protein isolate samples had not been inactivated.”
The PCR on UV-inactivated supernatant revealed very low copy number from only one cell line, which we believe supports our recommendation to UV-inactivate supernatant despite our observations of no cytopathic effect when supernatant that did not undergo UV exposure was tested on Vero cells.